# COVID-19, Mucormycosis and Cancer: The Triple Threat—Hypothesis or Reality?

**DOI:** 10.3390/jpm12071119

**Published:** 2022-07-10

**Authors:** Ishika Mahajan, Aruni Ghose, Deepika Gupta, Manasi Manasvi, Saisha Bhandari, Aparimita Das, Elisabet Sanchez, Stergios Boussios

**Affiliations:** 1Department of Medical Oncology, Apollo Cancer Centre, Chennai 600001, India; ishikaishan16@gmail.com; 2Department of Medical Oncology, Barts Cancer Centre, St. Bartholomew’s Hospital, Barts Health NHS Trust, London KT1 2EE, UK; aruni.ghose@nhs.net; 3Department of Medical Oncology, Mount Vernon Cancer Centre, East and North Hertfordshire NHS Trust, London KT1 2EE, UK; 4Department of Medical Oncology, Medway NHS Foundation Trust, Windmill Road, Gillingham ME7 5NY, UK; elizabet.sanchez@nhs.net; 5Division of Research, Academics and Cancer Control, Saroj Gupta Cancer Centre and Research Institute, Kolkata 700001, India; 6Microbiology, College of Medicine and Sagore Dutta Hospital, Kolkata 700001, India; guptadeepika.587@gmail.com; 7Internal Medicine, Kasturba Medical College, Mangalore 574142, India; manasimanasvi01@gmail.com; 8Internal Medicine, Post Graduate Institute of Medical Education and Research, Chandigarh 133301, India; saisha393@gmail.com; 9Faculty of Allied Health Sciences, Chettinad Academy of Research and Education, Chennai 600001, India; k.aparimita@gmail.com; 10Faculty of Life Sciences & Medicine, School of Cancer & Pharmaceutical Sciences, King’s College London, London SE1 9RT, UK; 11AELIA Organization, 9th Km Thessaloniki-Thermi, 57001 Thessaloniki, Greece

**Keywords:** cancer, mucormycosis, COVID-19, COVID-associated mucormycosis, diabetes, steroids, immunosuppression

## Abstract

COVID-19 has been responsible for widespread morbidity and mortality worldwide. Invasive mucormycosis has death rates scaling 80%. India, one of the countries hit worst by the pandemic, is also a hotbed with the highest death rates for mucormycosis. Cancer, a ubiquitously present menace, also contributes to higher case fatality rates. All three entities studied here are individual, massive healthcare threats. The danger of one disease predisposing to the other, the poor performance status of patients with all three diseases, the impact of therapeutics for one disease on the pathology and therapy of the others all warrant physicians having a better understanding of the interplay. This is imperative so as to effectively establish control over the individual patient and population health. It is important to understand the interactions to effectively manage all three entities together to reduce overall morbidity. In this review article, we search for an inter-relationship between the COVID-19 pandemic, emerging mucormycosis, and the global giant, cancer.

## 1. Introduction

The Severe Acute Respiratory Syndrome Coronavirus-2 (SARS-CoV-2)-induced Coronavirus disease 2019 (COVID-19) pandemic has witnessed more than 545 million illnesses and 6.3 million fatalities worldwide. It affected all national healthcare systems at different levels [1,2]. The southeast Asian belt constitutes the world’s second largest COVID-affected area. Second only to the USA, India boasts numbers larger than 43 million cases and 0.52 million deaths [2].

Bacterial and fungal co-infection alongside COVID-19 had been nil or minimal in Middle Eastern Respiratory Syndrome Coronavirus (MERS-CoV) and SARS-CoV-1 [1]. However, Huang et al. detected co-infection rates of 8 to 15% in COVID-19 patients [3]. With regard to fungi, the rate was around 3 to 6% [4]. The above invasive fungal infections (IFIs) include invasive aspergillosis, mucormycosis, candidiasis, and cryptococcosis [5].

Invasive Mucormycosis (IM), since the advent of the 21st century, has been deemed an emerging, angio-invasive fungal infection with drastically high mortality, of around 80% [6,7,8]. Historically speaking, India has been the principal hotbed of mucormycosis, recording a prevalence of 140 cases per million population. Globally, it is the highest and is eighty times higher than that of the affluent nations [6]. Hailing from the order *Mucorales,* Rhizopus is the most common species in the world. Apart from that, *Mucor* and *Lichtheimia* are the predominant causative agents in Europe, whereas *Apophysomyces* is the Asian, primarily Indian, counterpart [6,9]. The clinical spectrum includes five forms, primarily rhinocerebral, pulmonary, cutaneous, gastrointestinal, and disseminated disease [9]. Uncontrolled hyperglycaemia, steroid and immunomodulatory therapy, haematological malignancy (HMs) and haematopoietic stem cell transplantation (HSCT), solid organ malignancy (SOM) and transplant (SOT), persistent neutropenia, and iron overload are among the most notorious risk factors of IM [7,9].

In high-income countries (HIC) such as Europe, the USA, and Australia, HMs are the most common underlying cause of IM (38–62%) [7,9,10]. Acute myeloid leukaemia (AML) and acute lymphoblastic leukaemia (ALL), which frequently necessitate HSCT, are prominent examples [9,10]. These cases are at increased risk of IM, especially in their neutropenic state. Additionally, in these above patients, antifungal prophylaxis with azoles and echinocandins predisposes to breakthrough mucormycosis—voriconazole (52%), fluconazole (25%), caspofungin (9.8%), itraconazole (7.6%), and posaconazole (5%) [7,10].

Meanwhile, in low-and-middle-income countries (LMICs) such as the Indian subcontinent, diabetes mellitus (DM) is the most prevalent predisposing factor to IM [6,7,9]. Patel et al. identified rhino-orbito-cerebral mucormycosis (ROCM), constituting 65.7% of their IM cohort, to be the most common form [8]. The diabetic population of India is on the rise, from 60 million in 2011 to around 100 million by 2030 [11]. This risk relationship with IM is directly proportional. Prakash et al. designed a multi-centre study in India where 56.8% of IM patients were diabetics and 18% had diabetic ketoacidosis (DKA). Approximately 65% of them manifested ROCM [6].

COVID-19 patients have a higher prevalence of DM and DKA than that of the general population [12]. Compared to the other IFIs, mucormycosis in COVID-19 has proven to be a better-documented disease. A systematic review by John et al. reported the incidence of COVID-associated Mucormycosis (CAM) to be 87% [13]. Until 13 May 2021, Singh et al. reported that out of 101 CAM cases, 82 were from India and the remaining 19 were from the rest of the world [14]. From then until July, around 42,000 cases and 4000 deaths have occurred as a result of CAM in the Indian population [15].

The prevalence of COVID-19 among patients with cancer was higher in HMs (10.9%) as compared to SOMs (7.8%). Among SOMs, prostate cancer showed the highest frequency (10%). Patients receiving cancer treatment within 6 months (6.2%) were less susceptible to COVID-19 than those exceeding the above time interval (7.8%). COVID-19 patients with cancer have higher morbidity and mortality rates as compared to their negative counterparts—hospitalizations (43.8% vs. 31.5%), intensive care unit admissions (19.7% vs. 7.8%), respiratory support (7.9% vs. 1.3%), and deaths (14% vs. 3.1%). Among these COVID-19-positive cancer patients, mortality was lower in those treated with immunotherapy (7.1%) as compared to chemotherapy (14%), hormone therapy (16.2%), or even targeted therapy (14.1%) [16]. Among the 30.6% COVID-19 cancer patients enrolled in the UK Coronavirus Cancer Monitoring Project (UKCCMP) who died, fatality rates were higher in HMs as compared to SOMs. Leukaemia had the highest case fatality rate of 2.25. These HM patients who received recent chemotherapy experienced higher death rates during COVID-19 hospital admissions (odds ratio (OR) 2.09) [17].

In this review article, we search for an inter-relationship between the COVID-19 pandemic, emerging mucormycosis, and cancer.

## 2. CAM: Clinical Features

Rhizopus, as already mentioned, is ubiquitously present in soil, manure and as moulds on food [18]. Transmission of the infection can occur by airborne fungal spores entering via the nasal, oral, and conjunctival mucosa or by ingesting contaminated food. It may also be a commensal on the skin and mucosa, thus precipitating in opportunistic infections among the immunocompromised [19].

As discussed before, ROCM accounts for about a third of the cases of mucormycosis, with a steep rise noted in the COVID-19 pandemic [20]. The second most common type is pulmonary mucormycosis (PM) [13,14,20,21,22]. ROCM and PM originate in the nose and paranasal sinuses. The hallmarks of the disease are vascular invasion, thrombosis, and tissue necrosis. It results in bony destruction in the walls of the sinuses, leading to orbital and cranial spread [20,23].

Smith and Kirchner’s diagnostic criteria for ROCM from 1958 still remain the gold standard for clinical diagnosis [14]. In an immunocompromised setting, a combination of the following features ought to raise suspicion of mucormycosis:Facial pain and blood-tinged nasal discharge on the same side;Soft periorbital or peri-nasal swelling with skin and mucosal discoloration progressing to induration;Conjunctival suffusion, eyelid ptosis, eyeball proptosis, and complete ophthalmoplegia;Necrotic, black turbinates appearing like clotted, crusted blood;Cranial nerve palsies.

The other non-specific features include fever, unilateral headache, sinus tenderness, puffiness of the face, loosening of teeth, and nasal congestion [24].

PM has non-specific symptoms such as fever, cough, chest pain, dyspnoea, and haemoptysis [24]. The symptoms appear innocuously similar to the underlying viral infection. Hence, the early signs are written off as COVID-19. Suspicion of PM should arise in COVID-19 patients when these symptoms persist, or new symptoms develop after the resolution of the viral disease. Strict monitoring for reappearance or prolonged persistence of symptoms in diabetics and those receiving steroids and immunosuppressive therapies should prompt evaluation for mucormycosis [13,14,20].

In the immunocompromised population, the invasive nature of the fungus quickly turns the disease lethal [13,14,21,25]. A mortality rate of 30–50% is seen in CAM [13,14,20,21].

## 3. CAM: Road to Diagnosis

The first line of investigation performed is an imaging study [20]. The modalities recommended are magnetic resonance imaging of the paranasal sinuses with cerebral contrast for ROCM and simple computerized tomography (CT) thorax for pulmonary disease.

For pathologies of the paranasal sinuses, the imaging preferred is a CT scan [26]. IM may show as a unilateral hypodense opacification of the sinuses. It has a limited role as the soft tissue changes, and critical features such as perineural spread and cavernous sinus involvement can be missed. Its use is limited to picking up bony destruction only. Magnetic resonance imaging (MRI) has higher sensitivity than CT (86% vs. 57–69%) and similar specificity (83% vs. 81%) for acute invasive fungal sinusitis [26,27]. It produces hypo- or isointense lesions in all sequences with variable enhancement on contrast. The invasion of the fungus along the soft tissue and fatty planes is delineated with an increased clarity by T1-weighted images, aiding in the staging of the disease [28]. Detection of neurological complications such as cavernous sinus thrombosis, meningitis, necrosis, and subtle perineural invasion requires the utilization of cerebral contrast material such as gadolinium [27,29].

In pulmonary disease, with concomitant COVID-19 infection, it is exceedingly difficult to differentiate due to the similarities in the lesions and the acute respiratory distress syndrome picture. CT thorax shows ground-glass opacities and infiltration. Some cases may show features of consolidation and cavitation [30]. Angio-invasion in the lungs may cause thrombosis of the pulmonary vessels, appearing as wedge-shaped infarcts.

The suspicion of mucormycosis is based on direct microscopic examination of a wet mount. Staining with comparative fluorescent fungal stain, calcofluor white, and Giemsa is also performed on the clinical specimens at the outset [20,31].

The standard practice for confirmation of the diagnosis is a biopsy from the infected area demonstrating the presence of broad, wide ribbon-like aseptate hyphae branching at right angles, on a background of necrotic debris on histopathology [19,20,32,33]. Stains commonly used are haematoxylin-eosin, periodic acid-Schiff, or Grocott–Gomori’s methenamine-silver stain (GMS) [20,31].

On culture, the presence of cotton white or grey-black colonies at 28–30 °C and 35–37 °C is characteristic. The media commonly employed are Sabouraud Dextrose Agar and Brain and Heart Infusion Agar. Morphological identification from the fungal culture can be carried out by microscopic examination or by DNA sequencing on the basis of barcodes 18s, ITS, and MALDI-TOF [5,34,35]. Isolation from a fungal culture may be performed for differentiation of genus and species, and for antimicrobial susceptibility testing [5,32]. The treatment, however, is independent of the genus and species and is of epidemiological importance [36].

Serological investigations such as galactomannan assays and 1,3-β-D glucan assays support a diagnosis of fungal infection [20,35]. These tests are, however, typically negative in a pure mucor infection [20]. The utility of using the negative test to rule out the disease is questionable, as the tests are positive in mixed fungal infections [34]. While, presently, there is no serological test available for the rapid diagnosis of mucormycosis, ELISA and lateral flow immunoassay (LFIA) are in the process of development [37].

The clinical features, though characteristic of a fungal infection, are unfortunately non-specific, and may be mistaken for aspergillosis. Further, there exists a degree of misidentification of the Mucorales for the more ubiquitous Aspergillus spp. on microscopy [20,25,38]. There is thus a challenge in obtaining a confirmation of the diagnosis without immunohistochemistry staining with monoclonal antibodies or DNA-PCR of the tissue section. Both of these modalities are not only expensive but also not readily available at the point of contact [20,39].

The utility of complete blood counts is not established, especially in the present cohort of COVID-19 patients, where the white cell picture is deranged as a result of the primary infection [30,32,36,40]. For any analysis into the association of the derangement of these parameters due to the secondary fungal infection, confounding variables of the primary disease will have to be considered. The data on this are insufficient to comment upon and require further study.

## 4. Treatment

### 4.1. Mucormycosis

As discussed above, the suspicion of IM in the immunocompromised must be treated as a medical emergency due to its lethality [20]. In the present cohort, individuals infected with SARS-CoV-2, receiving treatment with steroids and immunomodulatory drugs, fall into the aforementioned high-risk category. Hyperglycaemia, due to a primary diabetic status or as a consequence of steroid therapy, increases the risk of infection [8,13,14].

Early diagnosis can lead to early initiation of treatment for positive outcomes and decreased case fatality rates: 20% fatality in early diagnosis vs. 60% in delayed diagnosis. Treatment guidelines can be set up in a three-pronged approach involving the utilization of antifungal drugs, surgical debridement, and an effort in part to increase immunity levels [41].

Prompt diagnosis directs the initiation of antifungal therapy wherein the first-line treatment is 5–10 mg/kg of Liposomal Amphotericin B (LAmB). A higher dosage of 10 mg/kg/day is recommended for central nervous system (CNS) involvement. If LAmB is not available, amphotericin B lipid complex or amphotericin deoxycholate can be used, with caution because of their known nephrotoxicity, and should be subjected to stringent therapeutic drug monitoring [20,42,43].

Other broad-spectrum antifungals such as posaconazole and isavuconazole may be utilized in the initial induction therapy where the patient cannot tolerate amphotericin B. Although not studied in randomized trials, their efficacy has been suggested to be similar to amphotericin [44]. There is, however, a noticeable need for updating the medical guideline in terms of induction of these agents for the treatment of CAM and their role as prophylactic agents [45].

Although antifungal prophylaxis for mucormycosis has been used for diseases such as underlying haematological malignancies, it is not recommended in COVID-19 patients [20,46].

Studies on combination antifungal therapy, where amphotericin B is combined with posaconazole or other antifungals such as caspofungin, have been carried out, which have not been shown to enhance survival or be more effective than monotherapy (35% versus 39%, respectively), and their cost to benefit ratio needs to be further evaluated on a case-by-case basis [42,44,47,48,49].

Step-down therapy is initiated with posaconazole delayed-release tablets (300 mg every 12 h on the first day, then 300 mg once daily) or infusions, which is preferred over oral suspensions. Oral isavuconazole is also an acceptable alternative, with doses of 200 mg three times a day on day 1–2 and 200 mg once a day from day 3, which is continued for a minimum of 6 weeks [20,43,50].

Add-on aggressive surgical intervention paves the way for a favourable response, which can be seen on imaging and clinical stabilization, in a minimum of 4–6 weeks [20,42,43]. Surgical debridement is carried out with clean margins, which serves three purposes—controls the disease, obtains specimens for histopathology, and confirms the microbiological diagnosis [20]. In cases of ROCM, there is extensive surgical debridement of the affected craniofacial tissues and risk of orbital exenteration, which may be lifesaving and has been found to be helpful even in cases where the infection has spread intracranially, according to the stage of involvement. Retrobulbar injection of amphotericin B may be given before surgery, according to the stage of involvement [42,43]. Lobectomy has also been performed with success in cases of PM where the infection was localized to a single lobe. The debridement procedures might need to be repeated as and when required. Endoscopic sinus surgery with limited tissue removal has also been performed [20].

While immediate surgical debridement has been the convention for lung, Saraiya et al. found that, after response to antifungal therapy, delayed surgical debridement (10 days) showed better patient outcomes than conventional aggressive surgical management. All five patients where the new protocol was followed survived [47].

Other modalities of treatment include hyperbaric oxygen therapy, administration of cytokines concurrently with antifungals, and deferasirox, but their use is not routinely recommended in the treatment guidelines [20].

Even after aggressive therapy and immediate initiation of the treatment, the prognosis is still quite grim with multiple studies showing a 90-day mortality average of 50%, with multiple patients discontinuing treatment because of financial constraints [8,48,49].

### 4.2. CAM

Methylprednisolone and dexamethasone, the cornerstones of COVID-19 treatment, are known to cause immunosuppression [50]. The World Health Organization (WHO) and the National Institute of Health (NIH) have recommended judicious use of systemic corticosteroids in COVID-19, which is in cases with evidence of respiratory failure or with oxygen saturation below 93–94%. These drugs unfortunately are known to be the most common cause of drug-induced hyperglycaemia [46,51].

A widespread disruption of antibiotic stewardship programs was noted due to rampant and injudicious use of antimicrobial agents (AMAs). Increased exposure of patients to AMAs, due to fear of superadded bacterial and fungal infection on COVID pneumonia, poor infection control measures due to rapidly changing protocols and the resulting confusion, led to an upstroke of nosocomial superbug infections in the background of COVID itself [52].

Hyperglycaemia is the most common risk factor for developing mucormycosis. In CAM, there should be immediate induction of antifungal therapy along with COVID-19 treatment protocols, and control of hyperglycaemia. Diabetic ketoacidosis, if present, should be emergently managed along with the reversal of immunosuppression in these patients [13,24,43,46,49,53,54].

Tocilizumab is another immunomodulator that is being used in the treatment of the disease. Studies and guidelines have recommended against indiscriminatory use of tocilizumab, which targets the immune pathways. Its use is only recommended for severe refractory disease where inflammatory markers are increased (e.g., C-reactive protein, interleukin 6) and in the absence of active fungal or bacterial infections [20,55].

Hence, upon detection of CAM, steroid doses should be reduced and immunomodulating drugs such as tocilizumab should be discontinued for reversal of immunosuppression [43,46,51].

## 5. COVID-19, Mucormycosis, and Cancer: The Triple Threat

### 5.1. COVID and Mucormycosis

As explained above, high-risk factors for mucormycosis include DM and metabolic acidosis, steroid usage in COVID-19, neutropenia, elevated serum iron levels, immunomodulatory therapies, and concomitant chronic illness. An increased likelihood of fungal infection has also been noted with haematological malignancies, organ and bone marrow transplant recipients, steroid therapy, patients on maintenance haemodialysis and iron chelation, trauma and burn victims [22].

SARS-CoV-2-mediated immune dysregulation of the cytokine storm in the background of a compromised immune system predisposes to invasive fungal infections [56]. This hyper-inflammatory state is also noted as a consequence of some malignancies, autoimmune diseases, and immunosuppressive medications [57]. High inflammatory markers, interleukins, interferon-gamma, tumour necrosis factors and hyperferritinaemia produce a picture similar to secondary haemophagocytic lymphohistiocytosis [3,58]. This hypercytokinaemia of severe COVID-19 produces diffuse alveolar lung damage, microvascular thrombosis, hyaline membrane formation and fibrosis, leading to an acute respiratory distress picture [59,60,61]. This unregulated increase in inflammatory markers and acute phase reactants downregulates the CD4+ T lymphocyte count, causing lymphopenia, propagating a decrease in viral clearance [62,63]. Additionally, there is a diffuse systemic vasculitic endotheliitis and microvascular damage, causing multiorgan dysfunction [64,65].

The vascular endothelial and alveolar damage is due to angiotensinogen converting enzyme- 2 (ACE-2) receptor-mediated viral entry into these cells. Other tissues with higher expression of these receptors such as gastrointestinal mucosa and proximal tubular cells of the kidney show a greater localised inflammatory response [66,67]. Patients with already compromised systems, i.e., with cardiovascular comorbidities such as hypertension, severe dyslipidaemia, obesity, DM, and those with chronic kidney disease thus have poorer outcomes in the event of disease progression to a cytokine storm [68,69]. Cardiac infection-producing myocarditis and infarctions are similarly hypothesized [70].

Immunosuppressive therapies such as steroids are among the few interventions proven to blunt this unchecked inflammation and improve survival. The high rate of usage of steroids in the period of cytokine storm in these patients exponentially increases the risk of them acquiring the IFIs in the recovery period [13,26].

The association of steroid therapy with mucormycosis has been the most constant in the recent upstroke of cases. Steroids are one of the only therapies proven to consistently reduce mortality in COVID-19 patients [13,26]. They are considered an essential therapy for COVID-19 patients on supplemental oxygen therapy [71]. Their use, however, has a multitude of complications attached. Worsening hyperglycaemia or new-onset DM and immunosuppression are among the common mechanisms that predispose to angio-invasive mucormycosis. Corticosteroids are implicated in immune system destruction by impairing leukocyte chemotaxis and prevention of phagolysosome fusion [72]. This steroid-induced immune dysregulation is an added burden on the virus-induced lymphopenia and hypercytokinaemia, leaving the patient wide open to a host of super-added co-infections. This may be compounded by the fact that steroids are given as over-the-counter pills in India leading to rampant misuse of the drugs [14].

DM remains the maximally associated factor predisposing to mucormycosis, with a mortality of 46% [48]. SARS-CoV-2 has been seen to worsen the glycaemic profile in patients with DM through multiple mechanisms, though they are all not completely understood. Notably, the virus can infect the pancreatic islet cells causing damage to the β-cells, causing hypoinsulinaemia and worsening hyperglycaemia [22,73,74]. DM, by nature, causes a degree of endothelial dysfunction due to chronic inflammation, leading to microvascular complications. Direct viral invasion of the vascular lining, causing endothelial damage, is mediated by ACE-2 receptors. This results in apoptosis and pyroptosis of the endothelium predisposing to angio-invasive secondary infections [61].

Among diabetics, the risk multiplies in patients with diabetic ketoacidosis (DKA). The virus and the hyperimmune stress state themselves predispose to dysglycaemia and DKA, even in the absence of poorly controlled sugars [22,74]. Hyperglycaemia produces a state of secondary immune deficiency. The dysglycaemia and acidic pH in ketoacidosis results in phagocyte dysfunction and defective intracellular killing of the fungi. The low-grade chronic inflammation causes an impaired immune response—both innate and adaptive response to infections. This also paves the way for the cytokine storm. The acidotic serum pH of 6.88–7.3 in DKA is also conducive to florid fungal proliferation [75].

The endothelial dysfunction and cytokine storm are procoagulant states leading to thrombosis and tissue ischemia [59,61]. The ischemic necrosis prevents leukocyte chemotaxis and effective delivery of antifungal agents to the foci of infection. The concomitant endotheliitis that occurs as a result of oxidative damage causes adherence of the fungal elements to the vessel walls. These factors combined with the angio-invasive nature of the fungus are responsible for the haematogenous dissemination of the disease [76].

Iron is required for fungal hyphal growth and development. One of the main host defence mechanisms against mucormycosis is limiting the availability of free iron to the fungus by having it bind to forms such as transferrin, ferritin, and lactoferrin. The acidic pH in DKA displaces the iron from its bound forms in the human body, increasing its availability to the fungal elements. The presence of increased unbound iron plays a vital role in predisposing patients with DKA to developing the disease. Some Mucorales secrete high-affinity iron chelators or siderophores such as rhizoferrin to acquire the iron from the host cells. Those species without rhizoferrin utilize exogenous xenosiderophores such as deferoxamine that are administered in DKA patients to treat iron overload, in order to fulfil their iron requirements. This explains why patients who are on long-term iron chelation therapy are at elevated risk of mucormycosis. Severe COVID-19 disease is also a hyperferritinaemic state, due to interleukin-six-mediated ferritin synthesis and downregulation of iron transport. There is an intracellular accumulation of iron in the hepatocytes, eventually causing necrosis and release of the accumulated iron. The high serum iron forms a fertile ground for florid fungal proliferation [77,78].

Tocilizumab, an immunomodulatory monoclonal antibody that targets the deregulated interleukin-six pathway implicated in the COVID-19 cytokine storm, is associated with an increased risk of infections. It modulates and suppresses the immune system, leaving the body vulnerable to other pathogens such as bacteria and fungi [79,80].

Neutrophil and phagocyte activity are the main defences against fungal infection. Mononuclear and polymorphonuclear cells generate reactive oxygen species and defensin peptides that inhibit fungal growth. Patients with impaired neutrophil function (primary immune deficiencies) are thus at higher risk. These are unaffected in COVID-19 infection; thus, a primary cause–association relationship is unlikely [72].

### 5.2. Cancer and Mucormycosis

Invasive fungal infections are a common cause of morbidity in patients with HMs. Second only to aspergillosis, zygomycosis occurs as a result of a confluence of multiple risk factors in this population [10,81]. Among the malignancies, acute leukaemia and lymphomas are found to be the most commonly associated with mucormycosis [82,83]. Other risk factors in the subgroup included haematopoietic stem cell transplants, myelodysplastic syndromes, concomitant graft versus host disease and high-dose steroid therapy [84]. The pulmonary and disseminated forms of the disease appear to be more common with haematologic malignancies [83,84].

Management of patients with HMs involves profound immunosuppression due to highly cytotoxic drugs, immunomodulatory therapies, and long-term high-dose steroids. All of these treatments break down the entire immune system to a virtually negligible existence, leaving the patient vulnerable to a host of opportunistic pathogenic infections [85].

These patients are at substantial risk for granulocytopenia and febrile neutropenia due to the chemotherapeutic medications. With neutrophils being the primary defence against the fungal elements, reduced numbers in the circulation lower the body’s defence and immunity predisposing to infection [86]. These patients show extensive angio-invasion and hyphal elements, but inflammatory infiltrates are minimal. Conversely, non-neutropenic HM patients with steroid compromised immune systems have a far less invasive disease, but more inflammatory infiltration with polymorphonuclear leukocytes [87,88].

Febrile neutropenia patients are on concomitant broad-spectrum antibiotic therapy to prevent flare-ups of bacterial infections. This in conjunction with the primary cytotoxic immunosuppressive medications leave this cohort of patients at risk for fungal infections, especially in the absence of primary antifungal prophylaxis [89].

Steroids form an essential arm in the primary therapeutic regimens of some lymphomas and myelomas. They are additionally used as adjuvant therapy for pain management [90,91], reducing inflammation and oedema, and for symptomatic relief of refractory symptoms of obstructions and dyspnoea [92]. The presence of prolonged high-dose steroid therapy causes a drug-induced hyperglycaemic state and, consequently, DM and DKA, creating an environment conducive for fungal growth [93].

In patients planned for or having received stem-cell transplantation, the phase of rebuilding immunity can be a perilous one. The complex immune milieu in these patients that results from the cocktail of immunosuppressive therapies, antimicrobial prophylaxis, and graft-versus-host disease (GVHD) multiplies the risk of acquiring invasive fungal infections [88]. This risk does not appear to be ameliorated even with antifungal prophylaxis with some azole and echinocandin medications. Breakthrough mucor infections are not uncommon [94,95]. In the last few years, mortality due to breakthrough mucormycosis has reduced with the advent of prophylaxis regimens with posaconazole. Treatment of mucormycosis in the cohort with liposomal amphotericin B and isavuconazole has also led to improvements in morbidity and mortality statistics [6,44,48].

Patients receiving chemotherapeutic drugs such as folate antagonists, antipurines, and steroids tend to develop oral ulcerations in the oral cavity and gastrointestinal tract. This discontinuity in the mucosa can pave the way for disseminated fungal infection, beginning as mucocutaneous or gastrointestinal disease [96].

### 5.3. COVID-19 and Cancer

Patients with cancer and concomitant COVID-19 infection have a higher fatality rate (7.6%) as compared to those without cancer (3.8%), with risk doubling [72,97]. There is no clear evidence of any associative relation linking the virus to any modulatory oncogenic pathobiology. However, there are a few theories postulating reasons for this doubling in mortality and high rate of complications [98,99].

Cancer treatments significantly increased risk of COVID-19 infection in the form of radiation therapy, chemotherapy, stem cell transplantation, and chimeric-antigen-receptor T-cell (CAR-T) therapy. All of these therapies cause varying degrees of immunosuppression in the body [100]. These treatments may sometimes cause a suppression of prodrome symptoms such as fever, leading to a delay in diagnosis [101].

Cellular senescence, via its oncogenic mutations, is a proven risk factor for many cancers [102]. Increasing age is a determinant of various other comorbidities, such as DM and hypertension that worsen the outlook of COVID-19 patients [73,102,103,104]. Inflammaging in a setting of cancer leads to a progressive weakening of the body’s defence mechanisms and healing [102,105]. This immune decline predisposes them to COVID-19 and also paves the way for poorer outcomes [104]. Recent immunophenotyping studies also show the virus promotes age-induced immune and genetic dysregulation, and this accounts for the vulnerability of the geriatric population [106]. This is supported by the fact that paediatric oncology patients show a lower infection rate as compared to their adult contacts and caregivers (2.5% as compared to 14.7%) [107,108].

The SARS-CoV-2 virus enters host cells by using its spike protein (S), which binds to the ACE-2 receptors on human cells [109,110]. These receptors are highly populated on type 2 alveolar cells, leading to pulmonary inflammation. They are also present on other tissues, such as the endothelium of the blood vessels, smooth muscles of the gastrointestinal tract, heart, kidneys, and liver [111,112]. Interestingly, the expression of the ACE-2 receptors varies in patients with cancer, i.e., some cancers show an upregulation of the receptor on their cell surfaces. This suggests that oncology patients are more likely to be infected with the virus and also have a poorer prognosis [113,114].

Both cancer and COVID-19 share a few metabolic risk factors. Type 2 DM, obesity, and metabolic syndrome are amongst the widely known hazards implicated, notably, in liver, pancreatic and endometrial cancers [115,116]. The immunologic response in cancer depends on the T helper cells and cytotoxic T cells. This mechanism is impaired by DM [117].

Lymphopenia is noted in a subset of patients with advanced cancers such as pancreatic, breast, lymphomas, and melanomas [118,119]. The lymphocytes also control the progression and therapeutic responses in cancer [120]. Additionally, the immune response of the body, adaptive and innate systems alters the natural history of the malignancy. A cytokine storm is a systemic inflammatory response syndrome that can be triggered in cancer patients by a plethora of factors, such as CAR-T cell therapy, monoclonal antibodies, infections, and some chemotherapeutic drugs [57,121,122]. The virus itself also causes an immune dysregulation, as part of its core pathogenesis, resulting in hypercytokinaemia [3,58]. This double-edged immune response may influence the pathogenesis of both diseases. The therapeutic approaches used in oncology patients to combat this syndrome can be considered as options in COVID-19-induced cytokine storm [123,124].

Coagulopathy is another common link between cancer and COVID-19 infection. Thrombosis and bleeding-related complications cause significant morbidity and mortality in severe COVID-19 infection [125,126]. They are also significantly accountable for cancer-related deaths, due to the direct thrombolytic properties of tumour cells, hypercoagulable state resulting from immune dysregulation, and release of pro-coagulant mediators such as cysteine proteases [127]. A combination of both entities can compound these procoagulant effects [98,99,128,129].

## 6. Conclusions

Despite there being a higher risk of morbidity, complications, and mortality, there is still a paucity of data emphasising the need for stricter guidelines, prophylactic therapies, and management strategies in cancer patients with COVID and systemic fungal infections such as mucormycosis. Studies into the interactions of the virus with cancer cells and the virus with chemotherapeutic drugs are also required to better understand the pathogenesis of infection and the interplay of the human body with cancer and virus cells. The presence of invasive fungal infection in this setting only complicates the treatment due to it being a superadded infection in a doubly immunocompromised host.

For physicians to effectively control and manage the triple threat posed due to a poor general condition from the malignancy, a viral infection and an invasive systemic fungal infection, a thorough study into the interplay of physical factors, chemical and microbiological factors is the need of the hour. Moreover, not only do COVID-19 and mucormycosis pose a larger risk in cancer patients, the COVID-19 and mucormycosis status of a patient with an underlying malignancy may also have bearing on the cancer chemotherapy/immunotherapy or antibody therapy. Figure 1 illustrates the interwoven relationship between COVID-19, mucormycosis and cancer.

It is therefore increasingly important to understand the interactions of the three entities to effectively manage them all together to reduce the morbidity presented. We hope this will invite further research to build up an evidence base for this triple threat—COVID-19, mucormycosis and cancer.

## Figures and Tables

**Figure 1 jpm-12-01119-f001:**
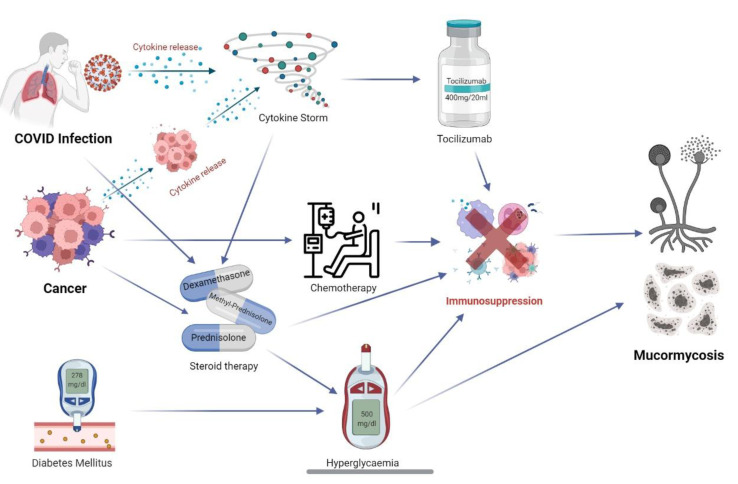
An illustration summarising the interwoven relationship between COVID-19, mucormycosis and cancer.

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
