# Peer review of "COVID-19, Mucormycosis and Cancer: The Triple Threat—Hypothesis or Reality?"

_jpm, 2022, doi:10.3390/jpm12071119_

Round 1

Reviewer 1 Report

It is a relevant article about COVID-associated mucormycosis, very interesting.

The main question of the research, is the connection between COVID-19 and mucormycosis or COVID-19 and cancer. This article adds more information about COVID-19. It shows the negative effect of steroids, which important against cytokine storm during COVID-19. The text is clear, The conclusions are consistent with the evidence, well written.

Author Response

Dear Editor and Reviewers,

I am pleased to resubmit for publication the revised version of the manuscript entitled “COVID-19, Mucormycosis and Cancer: The Triple Threat – Hypothesis or Reality?”.

Thankfully the reviewers provided us with a great deal of guidance, regarding how to better position the article. We are hopeful you agree that this revision will update our comprehensive review. All the comments have been addressed, as shown in the revised version of the manuscript, along with this point-by-point response to the reviewers' comments.

All corresponding are blue changes in the manuscript.

Reviewer #1:

It is a relevant article about COVID-associated mucormycosis, very interesting. The main question of the research is the connection between COVID-19 and mucormycosis or COVID-19 and cancer. This article adds more information about COVID-19. It shows the negative effect of steroids, which is important against cytokine storm during COVID-19. The text is clear, The conclusions are consistent with the evidence, well written.

Response:

Thank you for your comments. Your remarks are very encouraging.

Reviewer 2 Report

I read with great interest the paper. I find ti weel wrote and with good idea resarch

Below my minor suggestion

1. Introduction: updata data on SARS CoV2 wordwilde at the day of resubmission. 

2. CAM-clinical features: well wrote

3. treatment: add also the spread on antimicrobial resistance during COVID era. see Impact of SARS-CoV-2 Epidemic on Antimicrobial Resistance: A Literature Review. Viruses. 2021 Oct 20;13(11):2110. doi: 10.3390/v13112110. 

4. line 261: add the use of methylprednisolone and dexa only If there is respiratory failure or saturation under 93-94 

5. Figure 1. it is very well done

6. Conclusion: give some public health proposal that came from your interesting data

  1.  

  1.  

  1.  

Author Response

Reviewer #2:

I read with great interest the paper. I find it well written and with good idea research. Below my minor suggestion

1. Introduction: update data on SARS CoV2 worldwide at the day of resubmission.

2. CAM-clinical features: well wrote

3. treatment: add also the spread of antimicrobial resistance during COVID era. see Impact of SARS-CoV-2 Epidemic on Antimicrobial Resistance: A Literature Review. Viruses. 2021 Oct 20;13(11):2110. doi: 10.3390/v13112110.

4. line 261: add the use of methylprednisolone and dexa only If there is respiratory failure or saturation under 93-94

5. Figure 1. it is very well done

6. Conclusion: give some public health proposal that came from your interesting data

Response:

Your suggestions are much appreciated.

Point 1: Data modified, Kindly refer to lines 52-56 with reference no.2 updated to data as on 04/07/2022

The Severe Acute Respiratory Syndrome Coronavirus-2 (SARS-CoV-2) induced Coronavirus disease 2019 (COVID-19) pandemic has witnessed more than 545 million illnesses and 6.3 million fatalities worldwide. It affected all national healthcare systems at different levels [1,2]. The southeast Asian belt constitutes the world’s second largest COVID-affected area. Second only to the USA, India boasts of numbers larger than 43 million cases and 0.52 million deaths [2].

Reference:

2) WHO Coronavirus (COVID-19) Dashboard [Internet]. Covid19.who.int. 2022 [cited 4 July 2022]. Available from: https://covid19.who.int/.

Point 2, 5: Thank you.

Point 3: Addendum made, Please see lines 268-272 with change made to reference 52. (corresponding reference and citing changes made further).

A widespread disruption of antibiotic stewardship programs was noted due to rampant and injudicious use of antimicrobial agents (AMAs). Increased exposure of patients to AMAs, due to fear of superadded bacterial and fungal infection on COVID pneumonia, poor infection control measures due to rapidly changing protocols and the resulting confusion, lead to an upstroke of nosocomial superbug infections in the background of COVID itself [52].

Reference:

52) Segala F, Bavaro D, Di Gennaro F, Salvati F, Marotta C, Saracino A et al. Impact of SARS-CoV-2 Epidemic on Antimicrobial Resistance: A Literature Review. Viruses. 2021;13(11):2110.

Point 4: Note made at line 263-267, with change made to reference 51.

Methylprednisolone and dexamethasone, the cornerstones of COVID-19 treatment, are known to cause immunosuppression [50]. The World Health Organization (WHO), and the National Institute of Health (NIH) have recommended judicious use of systemic corticosteroids in COVID-19, that is in cases with evidence of respiratory failure or with oxygen saturation below 93-94%. These drugs unfortunately are known to be the most common cause of drug-induced hyperglycaemia. [46, 51].

51) COVID-19 Treatment Guidelines Panel. Coronavirus Disease 2019 (COVID-19) Treatment Guidelines. National Institutes of Health. Available at https://www.covid19treatmentguidelines.nih.gov/. [Accessed July 03, 2022].

Point 6:

We searched the literature for public health proposals and data linking COVID-19, Mucromycosis and Cancer, but were unable to find any. We believe that it is quite important the message of our paper that there is still a paucity of data emphasising the need for stricter guidelines, prophylactic therapies, and management strategies in cancer patients with COVID and systemic fungal infections like mucormycosis.